# Dietary Protein Intake and Transition between Frailty States in Octogenarians Living in New Zealand

**DOI:** 10.3390/nu13082843

**Published:** 2021-08-19

**Authors:** Ruth Teh, Nuno Mendonça, Marama Muru-Lanning, Sue MacDonell, Louise Robinson, Ngaire Kerse

**Affiliations:** 1Department of General Practice and Primary Care, University of Auckland, Auckland 1142, New Zealand; sue.macdonell@auckland.ac.nz (S.M.); n.kerse@auckland.ac.nz (N.K.); 2EpiDoC Unit, CEDOC, NOVA Medical School, Universidade Nova de Lisboa (UNL), 1099-085 Lisbon, Portugal; nuno.mendonca@nms.unl.pt; 3Comprehensive Health Research Centre (CHRC), NOVA Medical School, 1169-056 Lisbon, Portugal; 4James Henare Māori Research Centre, University of Auckland, Auckland 1142, New Zealand; m.murulanning@auckland.ac.nz; 5Population Health Sciences Institute, Newcastle University, Newcastle upon Tyne NE1 7RU, UK; a.l.robinson@newcastle.ac.uk

**Keywords:** frailty, mortality, protein deficiency, indigenous health, multi-state modelling

## Abstract

Adequate nutritional status may influence progression to frailty. The purpose of this study is to determine the prevalence of frailty and examine the relationship between dietary protein intake and the transition between frailty states and mortality in advanced age. We used data from a longitudinal cohort study of Māori (80–90 years) and non-Māori (85 years). Dietary assessments (24-h multiple pass dietary recalls) were completed at the second year of follow-up (wave 2 and forms the baseline in this study). Frailty was defined using the Fried Frailty criteria. Multi-state modelling examined the association of protein intake and transitions between frailty states and death over four years. Over three quarters of participants were pre-frail or frail at baseline (62% and 16%, respectively). Those who were frail had a higher co-morbidity (*p* < 0.05), where frailty state changed, 44% showed a worsening of frailty status (robust → pre-frail or pre-frail → frail). Those with higher protein intake (g/kg body weight/day) were less likely to transition from robust to pre-frail [Hazard Ratio (95% Confidence Interval): 0.28 (0.08–0.91)] but also from pre-frail to robust [0.24 (0.06–0.93)]. Increased protein intake was associated with lower risk of transitioning from pre-frailty to death [0.19 (0.04–0.80)], and this association was moderated by energy intake [0.22 (0.03–1.71)]. Higher protein intake in this sample of octogenarians was associated with both better and worse outcomes.

## 1. Introduction

Frailty is a multidimensional geriatric syndrome considered to be a state of increased vulnerability to external stressors (acute illness, injury, or stress) [1]. The inability to recover fully from these stressors is linked to adverse health outcomes over time, including an increased risk of falls, cognitive impairment, functional decline, re-admission to hospital and admission to residential age care [2]. Frailty can be defined using either a cumulative deficit approach or a phenotypic approach [3,4]. The frailty phenotype is most commonly determined by the presence of five physical characteristics: weakness, fatigue, slowness, unintentional weight loss and low physical activity [2], whereas cumulative deficit approaches, such as the Frailty Index, consider the accumulation of health conditions and impairments from physical, psychological and social domains [3,4]. Individuals in a prefrail or at-risk state (1–2 of these five characteristics) may or may not transition into overt frailty (three or more of the five) [5].

The prevalence of frailty increases with age [6], which, combined with the current worldwide growth in the number of older adults, will have an increasing impact on healthcare resources. Identifying factors that could form the basis of health interventions to reduce frailty onset or progression is, therefore, of major public health importance. Frailty often coexists with poor nutritional status [7] and is closely linked to one of the frailty phenotypes, unintentional weight loss; but also has a potential mechanistic link to weakness through loss of muscle mass

Dietary protein is required for muscle growth and repair. During digestion, protein is broken down into amino acids, the building blocks for muscle tissue. In addition to digestion, bioavailability of amino acids is also closely related to the absorption rates of amino acids [8]. With ageing, there is a higher rate of muscle protein breakdown and anabolic resistance which put older adults at increased risk muscle loss [8,9]. These mechanisms provide some explanation of previous epidemiological findings that lower dietary protein intakes are associated with poorer frailty status [10,11,12,13,14]. Few studies have investigated the association between dietary protein and the progression of frailty status in those aged 80 years and over, nor has the frailty status of older Māori people (indigenous people of Aotearoa New Zealand) been previously examined. Māori people have a different body composition with a higher ratio of lean mass to fat mass when compared to New Zealand European [15], at least in younger age groups, and it is not known how this difference in body composition is reflected in advanced age.

Thus, this study aimed to determine the prevalence of frailty in Māori and non-Māori in advanced age and to examine the relationship between dietary protein intake and the transition between frailty states and mortality in advanced age.

## 2. Materials and Methods 

### 2.1. The Life and Living to Advanced Age Cohort Study in New Zealand (LiLACS NZ)

Te Puawaitanga O Nga Tapuwae Kia ora Tonu, Life and Living to Advanced Age: A Cohort Study in New Zealand (LiLACS NZ) is a bicultural longitudinal study that aims to determine the predictors of successful advanced ageing. Multiple sampling strategies were used to contact eligible older adults, with the New Zealand (NZ) Māori Electoral Roll and NZ General Electoral Roll as the primary source. Eligibility criteria included non-Māori (mainly decedents from the United Kingdom) born in 1925 and Māori (the indigenous people of Aotearoa New Zealand) born between 1920 and 1930 who were living within defined regional boundaries in the central of the North Island. The wider age range for Māori was because of low population numbers in the age group and the aim to recruit equal size groups to allow meaningful analyses for Māori. Of the 1636 who were eligible, 937 octogenarians (57% response rate) were recruited at wave 1, of which nearly half (*n* = 421, 44%) were Māori. Details of the recruitment process have been described previously [16]. An annual comprehensive sociodemographic and health questionnaire was administered by trained interviewers between 2010 (wave 1) and 2016 (wave 6). In 2011 (wave 2) 62% (578/937) of the enrolled participants underwent detailed dietary assessment. Ethical approval was provided by the Northern X Regional Ethics Committee (NXT 09/09/088) in 2009, and all participants provided written consent.

To be eligible for inclusion in this sub-study (secondary analysis) participants needed to be living in the community at wave 2, and have complete frailty, body weight and protein intake data.

### 2.2. Dietary Protein Intake

Two 24-h Multiple Pass Dietary Recalls (24 h MPR) were completed on two different days by 62% of the original sample, 87% of wave 2 (216 Māori and 362 non-Māori) during the 2011 (wave 2, *n* = 660) data collection. Nutrient intakes were calculated using the New Zealand 2010 FOODfiles (Food Composition Database) [17]. Protein intake was expressed as gram per kilogram of body weight per day (g/kg BW/d, continuous), per increase in 10 g/d increments (continuous), and as a binary variable using cut points of ≥0.8 g/kg BW/d and ≥1.0 g/kg BW/d [9].

### 2.3. Anthropometry Functional and Health Measures

Body weight, kilogram (kg) was measured using the Tanita Inner Scan Body Composition Monitor, BC-545 (Tanita Corporation, Tokyo, Japan). Height, metre (m) was measured using a portable stadiometer, SECA 213 (SECA Corporation, Hamburg, Germany). For participants who were unable to stand, height was estimated from demi-span, a measure closely related to height [18]. BMI was calculated as weight/height^2^ (kg/m^2^). Grip strength was assessed using a Takei digital handgrip dynamometer-Grip D (Takei Scientific Instruments, Niigata City, Japan), over three attempts in each hand while in a standing position. The best performance by either hand was reported. Gait speed was assessed using the timed three-metre walk according to the protocol in the Short Physical Performance Battery [19] and expressed as metres per second (m/s). Physical activity level was assessed with the validated Physical Activity Scale for the Elderly (PASE) [20]. The PASE represents three domains of physical activities: recreational physical activities, housework related activities and work-related activities. Each activity was weighted according to different levels of intensity, and the score ranged from 0 to 793, with higher scores indicating greater physical activity. Co-morbidity was ascertained from self-reported chronic medical conditions and review of medical records from the general practices and hospital records from the Ministry of Health. Nineteen conditions were ascertained: coronary artery disease, cerebrovascular disease, congestive heart failure, peripheral vascular disease, hypertension, asthma/chronic obstructive pulmonary disease, osteoporosis, osteoarthritis, rheumatoid arthritis, diabetes, dementia, depression, cancer, thyroid disease, Parkinson disease, atrial fibrillation, anemia, renal impairment, and eye disease. A condition was considered present when reported in any of the data source(s). The total number of conditions was summed [21].

### 2.4. Frailty States

A frailty score was derived from each wave of data collection using the five phenotypes of the Fried Criteria [2]. Participants were assigned a score of 0 (absence) or 1 (presence) for each of five items; unintentional weight loss (≥5% weight loss in the last 12 months), weakness (grip strength < 20 kg for women; <30 kg for men) [22], poor endurance/fatigue (a negative response to “Do you feel full of energy?”), slowness (defined by gait speed < 0.8 m/s) [23], and low physical activity (sex-specific lowest quartile of the total PASE score). If gait speed was unable to be measured, the slowness criteria were imputed from an alternative question; “Do you walk around outside?” A negative response, or where assistance was required, was defined as the presence of slowness. The absence of all items scored 0 and was classified as robust, the presence of 1 or 2 items as pre-frail and 3–5 as frail [2].

#### Management of Missing Frailty Items

Where one or more frailty item was missing, participants could not be classified as robust. Those with missing items and who moved to residential aged care facility after wave 2, were classified as ‘frail’. In New Zealand, admission to residential age care facility follows a thorough needs assessment for people with physical, intellectual and/or sensory impairment or disability that reduces their ability to function independently over time. Community-dwelling participants with missing frailty items had frailty scores imputed based on the following criteria: one item missing and frailty score = 0–1, impute as ‘pre-frail’; one item missing and frailty score ≥ 2, impute as ‘frail’; two items missing and frailty score = 0, impute as ‘pre-frail’; and two or more items missing and frailty score ≥ 0, then impute as ‘frail’. Frailty was considered a missing value if all five items were missing (regardless of residence). This conservative imputation approach facilitates the retention of cases to improve the stability of the statistical models.

### 2.5. Statistical Analysis

Data from wave 2 forms baseline data for this study with frailty state transitions assessed between each wave, finishing with wave 6 of data collection. Descriptive statistics were completed for all variables. Categorical variables are reported as frequency (percentage, %). Variables with a normal distribution are presented as means and standard deviations (SD), while variables with a non-normal distribution are presented as medians and interquartile ranges (IQR).

Association between frailty status in Māori (adjusted for age) and non-Māori at baseline were assessed using ordinal regression model. Association between frailty status and potential confounders listed in Table 1 were assessed with chi-square test (categorical variables) or one-way ANOVA (continuous variables).

To determine the association of protein intake to transitions between frailty states and/or to death over four years, we fitted different multi-state models with increasing complexity (Model 1–3) and with different exposures of interest (increase of 10 g of protein/day, increase of 1 g of protein/kg of BW/day and ≥0.8 g/kg of BW/d). Model 1 is adjusted for age, ethnicity, sex and protein intake (and weight in the case of protein not expressed by body weight); Model 2 is further adjusted for co-morbidity; and Model 3 is further adjusted for energy intake. Due to the limited number of transitions between non-adjacent states (e.g., robust to frail) and the consequent non-convergence of the final models, we assumed that transitions from robust to frail or vice versa had to go through pre-frail, and that death from a robust state was not possible. Therefore, the allowed transitions were: Robust → Pre-Frail, Pre-Frail → Robust, Pre-Frail → Frail, Frail → Pre-Frail, Pre-Frail → Dead, and Frail → Dead.

Multi-state models describe the movement of an individual between finite states in a continuous time stochastic process under the Markov assumption that the next state is only influenced by the current state [26,27]. Multi-state models were fitted with the *msm* package in R v3.2.2 [28]. Point estimates and confidence intervals were used to assess statistical and clinical significance. The Broyden-Fletcher-Goldfarb-Shanno algorithm (quasi-newton optimization technique) was used to maximize the likelihood with results presented as hazard ratios (HR) and 95% confidence intervals (CI).

As a sensitivity analysis, models were further adjusted for misreporters, or education level (as categorized in Table 1), or using 1 g/kg BW/d or energy-adjusted protein intake (residual method) as the exposure, adding an interaction between protein intake and ethnicity. Models were fitted separately for Māori and non-Māori, or including participants living in rest homes or not imputing slowness. Misreporters were defined as having a ratio of energy intake: estimated basal metabolic rate below 1.05 and over 2.0 [25]. 

## 3. Results

Complete frailty data was available for 459 LiLACS NZ participants. One-third of participants (31.8%) were Māori with a mean (SD) age of 84.1 (2.6) years and Non-Māori 86.1 (0.5) years. There was a higher proportion of females in both Māori and non-Māori cohorts (57% and 53%, respectively). Participants without frailty scores at wave 2 differed from those in the analysis sample with a higher BMI, lower carbohydrate intake, and a greater proportion were Māori (Appendix A).

At baseline (wave 2), frail octogenarians had a higher co-morbidity (frail participants had 4.1 chronic diseases (SD:2.4), pre-frail 2.8 (SD:1.7) and robust 2.4 (SD:2.0) (*p* < 0.001)) (Table 1). While not statistically significant, frail participants tended to be older and had lower energy and protein intakes than robust and pre-frail participants Table 1 and Appendix A).

Fewer than one quarter of participants (22%, 102/459) were classified as robust (Fried score = 0) at baseline, with the majority (62%, 285/459) classified as pre-frail (Fried score = 1 or 2) (Figure 1). The percentage of participants classified as robust decreased yearly over the four years follow-up (e.g., 22% at baseline to 14% at 48 months). Conversely, the proportion classified as pre-frail was virtually stable (62% at baseline vs. 65% at 48 months). A significantly higher proportion of men than women were robust at 12-month (*p* = 0.005), 24-month (*p* = 0.003), and 48-month (*p* = 0.011) independent of age and ethnicity. Frailty did not differ between ethnic groups at any wave. 

In the Māori sample, the proportion of Māori men who were robust was relatively stable over the follow-up duration except for a dip at 36 months; a higher proportion of Māori women than men were frail (12 m *p* = 0.015; 24 m *p* = 0.006) (Figure 1b) In non-Māori participants, the difference in frailty status between men and women was comparable except at 24 and 48 months follow-up, a higher proportion women were frail compared to men (27% vs. 17—19%), but this difference did not reach statistical significance (Figure 1C).

### 3.1. Number of Transitions and Baseline Characteristics by Frailty State and to Death

A total of 1269 “frailty state transitions” were examined over four years of frailty data collection (Table 2). Of these, 692 remained unchanged in the same state from one wave to another. A further, 28 transitions to non-adjacent states (robust → frail, robust → death and recovery from frail → robust) were not considered due to the low numbers of each transition (Table 2). The total number of transitions used in the analytic sample, therefore, was 549 where 44% were from robust → pre-frail (124) or pre-frail → frail (117) (Table 2). These categories represent having at least one transition, and some participants had the same transition twice (therefore it does not sum to 549). Further, each participant may have had more than one different transition and, therefore, observations are not independent.

The distribution of participant characteristics at baseline (wave 2) for all allowed transitions are shown in Appendix A. Fewer women recovered from pre-frail to robust and/or transitioned from pre-frail to death. Moving from a pre-frail state to death was the most common transition for Māori (39% vs. approximately 20% for robust to pre-frail (Appendix A)). Those who recovered from pre-frail to robust had lower disease burden and, in general, higher nutritional intake (energy, protein, carbohydrate and fat).

### 3.2. Protein Intake and Transitions between Frailty States and to Death

We fitted different multi-state models with increasing complexity (Model 1–3) and with different exposures of interest (increase of 10 g of protein/d, intake of 1 g/kg of BW/d and ≥0.8 g/kg of BW/d) (Table 3). There were no statistically significant differences in the likelihood of transition with lower or higher protein intake (≥0.8 g/kg BW/day) (Table 3 and Appendix A). Those with higher protein intake (per 1 g/kg of BW/d were less likely to transition from robust to pre-frail (Model 3: g/kg BW/d: HR: 0.28, 95%CI: 0.08–0.91) and pre-frail to dead (Model 2: g/kg BW/d: HR: 0.19, 95%CI: 0.04–0.80). Unexpectedly, those with higher protein intake (g/kg BW/d) tended to also be less likely to recover from pre-frail to robust (Model 3: g/kg BW/d: HR: 0.24, 95%CI: 0.06–0.93) (Table 3). Conclusions did not differ in any of the sensitivity analyses.

## 4. Discussion

There was a high prevalence of pre-frailty and frailty in New Zealand adults of advanced age in LiLACS NZ. Frailty at baseline was not associated with protein intake but was associated with a higher co-morbidity, and the proportion of participants classified as frail increased as the cohort aged. The trends of frailty prevalence observed in LiLACS NZ were similar to those seen in previous studies of community-dwelling people in advanced age [29,30,31,32]. For instance, while fewer participants in LiLACS NZ were frail at baseline compared to those in the Newcastle 85+ study, (16% vs. 24%, respectively) [29], prevalence in frailty status were similar over time. Unlike the Newcastle 85+, we did not see any significant differences between men and women in New Zealand. However, co-morbidity was a characteristic of frail participants in both studies.

This is the first specific reporting of frailty status and transitions for Māori in advanced age, to our knowledge. Māori participants had a lower prevalence of frailty at all waves of the study, although this did not reach statistical significance. This tendency may reflect the younger age of Māori participants and that they had a greater level of physical activity at baseline [33]. It is notable, however, that Māori participants were more likely to have chronic health conditions that may ultimately contribute to the development of frailty including diabetes, respiratory conditions, and congestive heart failure [33]. 

In our sample, nearly one-fifth (19%) of the transitions between frailty states signified a deterioration in frailty state (e.g., from robust to prefrail or pre-frail to frail) and more than half (54%) of the transitions remained the same. Frailty transition states were not unidimensionally related to protein intake. Of the three measures of protein intake, only one measure was associated with frailty transition, i.e., the amount of protein intake by gram per kilogram (kg) body weight (BW) per day. Participants with higher protein intake were less likely to transition from robust to pre-frail or pre-frail to death over four years in models adjusted for key covariates. What is surprising was that participants with higher protein intake (each gram increase in protein intake per kg of body weight) were less likely to recover from pre-frail to robust. Our findings suggest a higher protein intake may halt the deterioration of frailty status but perhaps not reverse the frailty trajectory. In addition to the quantity of protein consumed, protein quality and distribution coupled with temporal physical activity may reverse the frailty trajectory [9]. 

In cross-sectional studies, animal protein intake was found to have a positive dose-response association with muscle mass in healthy older women (mean age 66 years old) but not for plant protein [34]. However, the beneficial effect of animal protein was not observed in frailty status, a measure of geriatric syndrome. In the Rotterdam Study (mean age 75 years old), animal protein was not associated with risk of frailty. What is interesting in the Rotterdam Study was an inverse association between plant protein and risk to be frail, i.e., a higher plant protein was associated with a lower risk to be frail when compared to the robust group [35]. In LiLACS NZ, the protein source varied between sex and ethnic groups. Overall, there was a higher proportion of animal protein than plant protein (approx. 57% vs. 43%) [36], a proportion similar to the Rotterdam Study [35]. The impact of protein source and distribution on transitions of frailty states was beyond the scope of this paper, and further examination is warranted. 

Our study also showed the association between protein intake and transitioning from pre-frail or frail to dead was in an opposite direction. Higher protein intake was less likely to transition from pre-frail to dead but more likely to transition from frail to dead. These associations were, however, moderated by energy intake. Reverse causality between energy intake in frail older adults may partly explain protein intake was not as strong in this sample of octogenarians than in other studies. We note that the HR for the significant transitions had a wide CI possibly due to the low number of transitions and/or distribution of protein intake.

Protein intake was hypothesized to have an impact on frailty transition due to its role in muscle anabolism. Cross-sectional studies have found that a higher protein intake was inversely associated with frailty status in septuagenarian but not in octogenarians [11]. In longitudinal studies, low protein intake was associated with increased odds of frailty in sexagenarians but not in septuagenarians [11]. Protein intake in these observational studies were expressed by various methods and different cut points: dichotomous groups (based on recommended dietary protein intake cut points 0.8 g/kg BW/day), quintiles, tertiles, or total daily protein intake. Our findings were in line with previous studies when protein intake was expressed as high vs low (≥0.8 or ≥1.0 g/kg BW/day). The null association between frailty status and protein intake (when expressed by as high vs. low) is potentially associated with intrinsic factors, such as age-related changes in protein metabolism and other factors including immunosenescence. Changes in the gastrointestinal tract alter proteolysis, thereby affecting protein digestion and absorption [9,37]. Anabolic resistance that has been noted in older adults also contributes to the imbalance between muscle protein synthesis and muscle protein breakdown [9]. In view of these biological changes, a higher protein intake may be required to halt the deterioration of frailty status. In our sample, both men and women met the EAR for protein intake but not the RDI, although the robust group come close to the recommendation. Achieving the current RDI may not be sufficient to ward off lean muscle mass loss, particularly for those living to advanced age. Progression to frailty occurs along a continuum, which provides opportunities for targeted interventions to slow or halt the progression to full frailty [5].

We observed a significant association between the number of co-morbidities and frailty, similar to that observed in previous work [29]. A recent meta-analysis of observational studies conducted with community-dwelling older adults found that multimorbidity was associated with more than a two-fold increased likelihood of being frail (odds ratio = 2.27; 95% CI = 1.97–2.62) [38]. Both the English Longitudinal Study of Ageing (aged ≥ 75 years) [39] and the Newcastle 85+ Cohort Study [29] also found that chronic disease was positively associated with frailty progression. These findings emphasize the complex interplay of multimorbidity and functional disability that contribute to frailty, as examined by Pivetta et al. using path analysis [31].

### Strengths and Limitations

This study included detailed dietary protein intake and annual measurement of frailty status of octogenarians living in New Zealand and, to our knowledge, is the first paper to report on the association between protein intake and frailty status in older Māori. Several limitations are acknowledged. Dietary assessment (and protein intake) was only available at wave 2 and, therefore, assumed to remain stable or to change proportionally in all participants. Dietary intake is largely determined by our eating behavior which is quite stable throughout the life-course. Probable factors impacting eating habits in this age group include, but are not limited to, changes in living situation, health status, and medication used [40]. We could not assign a frailty score for 139 (20%) participants at baseline, and it is possible that those not included in the analytic sample were frailer than those who were. We were not able to further adjust the various models for physical activity level in this analysis as the activity level was included in the frailty score. Separate analysis of frailty transitions was not possible for Māori participants due to the limited number of transition states. However, this was addressed by adjusting for ethnicity in the model. Multi-state modelling follows the Markov assumption that the next (frailty) state is only influenced by the current state and, therefore, history of frailty status (prior to previous state) is not considered. We recommend cautious interpretation of these findings in the context of the study limitations and the wide confidence intervals for some exposures attributed to the small sample size.

## 5. Conclusions

In this sample, nearly two-thirds of octogenarians were pre-frail and one-fifth were robust. There was a higher proportion of older women than men who were frail over four years. Octogenarians with higher protein intake were less likely to transition from robust to pre-frail and pre-frail to dead. However, they were also less likely to recover from pre-frail to robust. These findings suggest that the amount of protein intake alone was insufficient to reverse the frailty trajectory but was able to prevent deterioration of frailty states. Future studies integrating multiple approaches (quality and distribution of dietary protein intake) with physical activity are warranted to maintain optimal function in older adults transitioning to advanced age.

## Figures and Tables

**Figure 1 nutrients-13-02843-f001:**
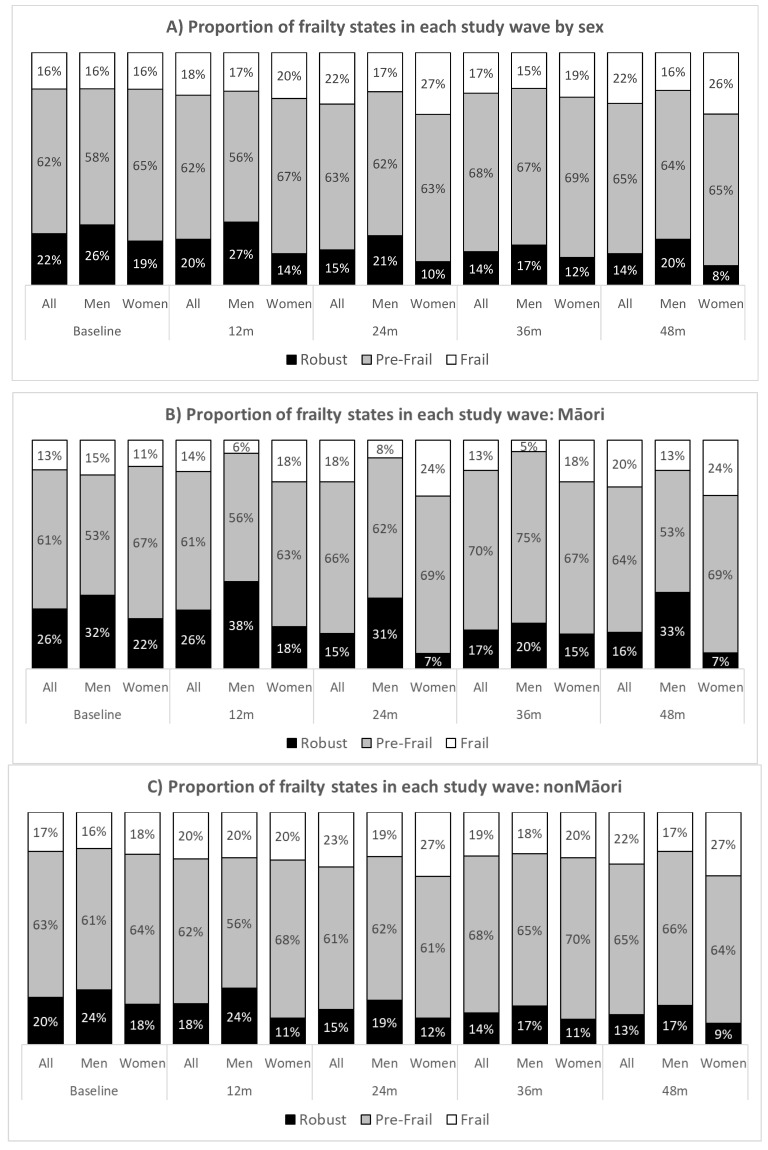
Frailty states by sex (**A**) and ethnicity: Māori (**B**) non-Māori (**C**) for all five waves.

**Table 1 nutrients-13-02843-t001:** Health and sociodemographic characteristics of 459 robust, pre-frail and frail LiLACS NZ participants at baseline (wave 2).

	Robust Fried Score = 0	Pre-Frai Fried Score = 1–2	Frail Fried Score = 3–5	Total	*p* *
*n*	102 (22%)	285 (62%)	72 (16%)	459	
Women, *n* (%)	47 (46.1)	161 (56.5)	39 (54.2)	247 (53.8)	0.194
Age, (years)	85.1 (2.0)	85.4 (1.7)	85.7 (1.7)	85.4 (1.8)	0.067
Māori, *n* (%)	38 (37.3)	89 (31.2)	19 (26.4)	146 (31.8)	0.299
Education, *n* (%)					0.956
Primary or no schooling	17 (16.8)	62 (21.8)	19 (26.4)	98 (21.4)	
Secondary school, no qualification	37 (36.6)	101 (35.6)	23 (31.9)	161 (35.2)	
Secondary school, qualification	22 (21.8)	59 (20.8)	15 (20.8)	96 (21.0)	
Trade, occupational	9 (8.9)	24 (8.5)	6 (8.3)	39 (8.5)	
Tertiary qualification	16 (15.8)	38 (13.4)	9 (12.5)	63 (13.8	
Co-morbidity	2.4 (2.0)	2.8 (1.7)	4.1 (2.4)	2.9 (2.0)	<0.001
Body weight, (kg)	73.4 (11.1)	70.7 (13.1)	71.2 (16.3)	71.4 (13.3)	0.219
BMI, (kg/m^2^)	27.7 (4.0)	27.1 (4.3)	27.3 (6.4)	27.3 (4.7)	0.555
Energy intake, (MJ/d)	7.5 (2.4)	7.1 (2.2)	6.8 (2.4)	7.2 (2.3)	0.141
Protein intake, (g/d)	71.1 (24.1)	67.3 (21.9)	64.1 (25.7)	67.7 (23.1)	0.134
Men	80.8 (25.2)	74.5 (23.2)	75.6 (28.3)	76.3 (25.6)	0.288
Women	59.8 (17.0)	61.8 (19.1)	54.4 (18.7)	60.3 (18.8)	0.086
Protein intake, (g/kg BW/d)	0.99 (0.37)	0.97 (0.31)	0.93 (0.38)	0.97 (0.34)	0.508
Men	1.08 (0.39)	0.97 (0.32)	1.02 (0.43)	1.01 (0.36)	0.177
Women	0.89 (0.32)	0.97 (0.31)	0.86 (0.32)	0.93 (0.31)	0.071
≥0.8 g/kg BW/d, *n* (%)	66 (64.7)	198 (69.5)	44 (61.1)	308 (67.1)	0.339
Men	42 (76.4)	89 (71.8)	24 (72.7)	155 (73.1)	0.814
Women	24 (51.1)	109 (67.7)	20 (51.3)	153 (61.9)	0.039
≥1.0 g/kg BW/d, *n* (%)	42 (41.2)	118 (41.4)	26 (36.1)	186 (40.5)	0.708
Men	28 (50.9)	49 (39.5)	15 (45.5)	92 (43.4)	0.353
Women	14 (29.8)	69 (42.9)	11 (28.2)	94 (38.1)	0.103
Carbohydrate intake, (g/d)	194 (58.6)	187 (55.0)	178 (60.0)	187 (56.7)	0.203
Fat intake, (g/d)	75.5 (32.7)	71.4 (31.9)	69.7 (37.1)	72.1 (33.0)	0.455
Misreporters, *n* (%)	30 (29.4)	86 (30.2)	26 (36.1)	142 (30.9)	0.580

Entries are means ± SD unless mentioned. * Unadjusted non-difference between frailty states by chi-square test (categorical variables) or one-way ANOVA (continuous variables). Estimated average requirement (EAR) and recommended dietary intakes (RDI) for adults > 70 years in Australia and New Zealand: Men: 65 g/day and 81 g/day. Women: 46 g/day and 57 g/day [24]. The EAR and RDI for adults > 70 years in Australia and New Zealand: Men: 0.86 g/kg BW/day and 1.07 g/kg BW/day. Women: 0.75 g/kg BW/day and 0.94 g/kg BW/day [25]. Co-morbidity (measured at wave 1) is the sum of 19 chronic diseases. All other variables were measured one year later at wave 2 (considered baseline in this study). BMI, body mass index; BW, body weight; MJ, megajoules.

**Table 2 nutrients-13-02843-t002:** Number of “transitions” between each frailty state and to death over four years.

	To	Robust	Pre-Frail	Frail	Dead
**From**					
Robust		95	124	4	20
Pre-frail		84	501	117	87
Frail		4	74	96	63

For example, 124 transitions were from robust to pre-frail. Notes: Font color red: worsening of frailty transitions; grey: unchanged; green: improved; black: transitions between non-adjacent states (e.g., robust to frail). Robust, Fried score = 0; Pre-frail, Fried score = 1–2; Frail= Fried score = 3–5.

**Table 3 nutrients-13-02843-t003:** Hazard ratios and 95% confidence intervals for the association between protein intake and transitions between frailty states and to death over four years.

	Increase of 10 g/day		Increase of 1 g/kg BW/day		≥0.8 g/kg BW/day	
	HR	95% CI	HR	95% CI	HR	95% CI
Robust → Pre-Frail (*n* = 124)						
Model 1	0.95	0.85–1.06	0.49	0.25–0.96	0.74	0.47–1.17
Model 2	0.94	0.84–1.06	0.50	0.25–1.02	0.75	0.47–1.21
Model 3	0.85	0.69- 1.05	0.28	0.08–0.91	0.74	0.43–1.25
Pre-Frail → Robust (*n* = 84)						
Model 1	0.92	0.82–1.04	0.49	0.22–1.09	0.60	0.33–1.10
Model 2	0.92	0.81–1.03	0.47	0.21–1.05	0.61	0.33–1.13
Model 3	0.81	0.65–1.01	0.24	0.06–0.93	0.58	0.30–1.14
Pre-Frail → Frail (*n* = 117)						
Model 1	1.00	0.91–1.09	1.02	0.57–1.80	0.71	0.47–1.07
Model 2	1.00	0.91–1.11	1.19	0.66–2.15	0.72	0.46–1.13
Model 3	1.03	0.87–1.22	1.59	0.72–3.55	0.65	0.39–1.09
Frail → Pre-Frail (*n* = 74)						
Model 1	1.06	0.95–1.19	1.59	0.76–3.32	0.92	0.49–1.72
Model 2	1.06	0.94–1.19	1.63	0.76–3.52	0.77	0.39–1.55
Model 3	1.04	0.87–1.24	1.78	0.63–5.05	0.66	0.30–1.43
Pre-Frail → Dead (*n* = 87)						
Model 1	0.85	0.68–1.06	0.29	0.05–0.92	0.72	0.30–1.70
Model 2	0.85	0.68–1.06	0.19	0.04–0.80	0.63	0.27–1.47
Model 3	0.98	0.65–1.49	0.22	0.03–1.71	1.16	0.39–3.47
Frail → Dead (*n* = 63)						
Model 1	1.07	0.97–1.19	1.96	1.05–3.66	0.93	0.56–1.56
Model 2	1.07	0.96–1.19	2.01	1.09–3.73	0.99	0.59–1.67
Model 3	1.03	0.87–1.22	2.04	0.94–4.45	0.76	0.42–1.40

Model 1 is adjusted for age at wave 2, sex and ethnicity, (and weight in the case of protein not expressed by body weight). Model 2 is further adjusted for co-morbidity and Model 3 is further adjusted for energy intake. BW, body weight; HR, hazard ratio; CI, confidence interval.

## Data Availability

LiLACS NZ data available on request contacting the principal investigator of the study Professor Ngaire Kerse.

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
