# Peer review of "Dietary Protein Intake and Transition between Frailty States in Octogenarians Living in New Zealand"

_nutrients, 2021, doi:10.3390/nu13082843_

Round 1
Reviewer 1 Report
Nutrients-1305094 presents results for the association between protein intake and frailty transitions. While some parts of this manuscript were interesting, other areas could be improved. I hope the authors consider my feedback.
MAJOR COMMENTS
- Section 2.3: This section could be detailed a bit more. I understand the research was a secondary analysis, but what is listed here is important as it pertains to some of the frailty components.
- Section 2.4.1: Treating missing frailty items here is largely based on assumption and artificial inflation of frailty cases is likely occurring. Strong justification in the text is needed to support this method.
- Line 134: This is the first-time death was mentioned. It is out of place.
- Section 2.5: Just present the results of the fully-adjusted models. Examining how covariates attenuated the estimates was not part of the purpose.
- Stat Analysis: You may want to conduct the all analyses by Maori and non-Maori, and include as an appendix where appropriate. It just seems odd that Figure 1 examines this demographic, but the other findings do not.
MINOR COMMENTS
- The title needs revision because it does not stand-alone (e.g., “LiLACS NZ”).
- Lines 40-43: Use caution here. There are multiple frailty definitions and what you are providing is a single example.
- Line 60: Please define “NZ” before abbreviating. Other abbreviations were not defined before usage elsewhere in the paper. Please correct.
- Table 1: It is challenging to know which groups are significantly different in this table.
- Figure 1A: Where are the waves? It might also be interesting to conduct a statistical test of inference on each of these data (i.e., Figure 1 A, B, C).
- Table 2: What do the colors mean?
- Correct minor English language edits throughout.
- Make any changes to the abstract that align with those from the text.
Reviewer 2 Report
Dear Authors,
Thank you for the opportunity to review your manuscript” Dietary protein intake and transition between frailty states in advanced age: LiLACS NZ”
Overall, it is interesting, however, the outcomes based on the assessment are potentially quite predictable.
Please review the comments below with the objective of substantiating the outcomes further.
Material and Methods:
Line 69: You state, recruited 937 octogenarians of which half (n=421, 69 44%) were Māori, the indigenous people of New Zealand.; Firstly, why were 937 people recruited? Is there a format to suggest that this will give you a confidence that you are within accurate level (Was a G Power analysis completed?). Can you also include the recruitment criteria acceptance and rejected. It may have been outlined previously, However, it needs to be obvious.
It would also be very useful as you have stated you conducted a dietary assessment to list within each group what the dietary protein was , the volume and how this compares to the New Zealand Nutritional requirements with respect to avoiding frailty.
Please also list the split between sex, age and ethnicity within the groups.
Line 142-143: You have stated; Therefore, the allowed transitions were: Robust → Pre-Frail, Pre-Frail → Robust, Pre-Frail → Frail, Frail → Pre-Frail, Pre-Frail → Dead and Frail → Dead.” It would be very helpful to understand the scores assigned to each one of these groupings so that the scales are clear on how you evaluated the participants.
Line 168: You state: Fewer than one quarter of participants (22%) were classified as robust at baseline, with the majority (62%) classified as pre-frail. Please explain how this was evaluated as it is not clear. It would be very helpful if you place the scored assessments next to your comments to quantify the statements.
Line 279 you state: “We observed a significant association between co-morbidity and frailty, similar to that observed in previous work.” Please explain the co-morbidities being referred to and how these are related to the frailty assessment. Please also state what the previous work was, by whom and when.
Based on the limitations in this study there is very little confidence in the outcomes as you have made substantial assumptions. Can you please explain these further if possible.
Round 2
Reviewer 1 Report
The authors have done a nice job addressing my previous concerns. I hope the authors consider some additional feedback for further improving their paper.
- Line 169: The classification of “dead and frail to dead” might be confusing for the reader. What is important instead here is “frail to dead”. Transitioning from dead to dead is irrelevant.
- Table 1: We do not know how the characteristics of the participants in the robust, pre-frail and frail groups differ. Only a p-value is listed in the far-right column. Specific differences are not identified. Consider using symbols to determine between groups differences or list them in the text.
Author Response
Please see the attachment.
- Line 169: The classification of “dead and frail to dead” might be confusing for the reader. What is important instead here is “frail to dead”. Transitioning from dead to dead is irrelevant.
Response: Thank you for picking this up, it is not possible to transition from dead to dead. We have amended the sentence, i.e. added the missing comma. “Therefore, the allowed transitions were: Robust → Pre-Frail, Pre-Frail → Robust, Pre-Frail → Frail, Frail → Pre-Frail, Pre-Frail → Dead, and Frail → Dead.”
- Table 1: We do not know how the characteristics of the participants in the robust, pre-frail and frail groups differ. Only a p-value is listed in the far-right column. Specific differences are not identified. Consider using symbols to determine between groups differences or list them in the text.
Response: We thank the reviewer for the suggestion. Although a formal statistical test was performed, Table 1 is intended as a descriptive analysis and not an inferential one. We believe that a post-hoc analysis would further increase the multi-testing issue. The hypothesis of the manuscript is tested by modelling the transitions between frailty states with multistate models and inferring if protein intake changes the transition probabilities.

Reviewer 2 Report
Dear Authors,
Thank you for the reply to the issued raised, I note that the answers provided did not address the issues for point 1-4 and also you have not explained what the co-morbidity's are as requested in point 5 and with respect to point 6 the confidence levels to accept the results submitted have not been ratified or explained.
Author Response
Response to Reviewer 2 additional comments
Original feedback: Thank you for the opportunity to review your manuscript” Dietary protein intake and transition between frailty states in advanced age: LiLACS NZ”
Overall, it is interesting, however, the outcomes based on the assessment are potentially quite predictable.
Please review the comments below with the objective of substantiating the outcomes further.
Thank you for your time to reviewing our manuscript and offering constructive comments. Please see below our responses.
Additional feedback: Thank you for the reply to the issued raised, I note that the answers provided did not address the issues for point 1-4 and also you have not explained what the co-morbidity's are as requested in point 5 and with respect to point 6 the confidence levels to accept the results submitted have not been ratified or explained. (response suffix with a)
Material and Methods:
Point 1 Line 69: You state, recruited 937 octogenarians of which half (n=421, 69 44%) were Māori, the indigenous people of New Zealand.; Firstly, why were 937 people recruited? Is there a format to suggest that this will give you a confidence that you are within accurate level (Was a G Power analysis completed?). Can you also include the recruitment criteria acceptance and rejected. It may have been outlined previously, However, it needs to be obvious.
Response 1: We have clarified this in the text. Line 96 mentioned that this sub-study (secondary analysis) include only those who were living in the community at wave 2.
Response 1a: As this is a secondary analysis, we did not complete a G Power analysis. Although very rare, power analysis of an observational study and secondary data may be important, but we have to run it for every transition and not only the difference between frailty states at baseline. That takes time and our statistician don’t think the reviewer’s comment warrants that.
Point 2 It would also be very useful as you have stated you conducted a dietary assessment to list within each group what the dietary protein was , the volume and how this compares to the New Zealand Nutritional requirements with respect to avoiding frailty.
Please also list the split between sex, age and ethnicity within the groups.
Response 2: We have expanded Table 1 with table notes stating the EAR and RDI for men and women above 70, and added a supplementary table (Table S2, Line 569-577)
Response 2a: In New Zealand, the recommended daily intake for men and women aged above 70 years old to sustain overall health are 81g/day and 57g/day respectively. The recommended relative protein intake for men is 1.07g/kg/day and women 0.94g/kg/day respectively. We added notes below Tables 1 and S2.
We have revised the sentence in the discussion “In our sample, both men and women met the EAR for protein intake but not the RDI, although the robust group come close to the recommendation. Achieving the current RDI may not be sufficient to ward off lean muscle mass loss [28] particularly for those living to advanced age.”
Also we have expanded Table S2 reporting the frequency and % of those meeting the EAR.
Point 3 Line 142-143: You have stated; Therefore, the allowed transitions were: Robust → Pre-Frail, Pre-Frail → Robust, Pre-Frail → Frail, Frail → Pre-Frail, Pre-Frail → Dead and Frail → Dead.” It would be very helpful to understand the scores assigned to each one of these groupings so that the scales are clear on how you evaluated the participants.
Response 3: Thank you for your comment. The definition of frailty status was mentioned in section 2.4 (line 142-143). The specific measurements used (e.g PASE) are described in section 2.3 Anthropometry functional and health measures (line 116-124). We have added the assessment score next to the comments (line 210-211)
Response 3a: Frailty score was assessed using the Fried cirteria. The absence of all five items scored 0 and was classified as robust, the presence of 1 or 2 items as pre-frail and 3-5 as frail. We have added scoring to Table 1
Point 4 Line 168: You state: Fewer than one quarter of participants (22%) were classified as robust at baseline, with the majority (62%) classified as pre-frail. Please explain how this was evaluated as it is not clear. It would be very helpful if you place the scored assessments next to your comments to quantify the statements.
Response 4: The definition of frailty status was mentioned in section 2.4 (line 142-143). We have added the assessment score next to the comments (line 210-211)
Response 4a: We have revised the sentence as followed “Fewer than one quarter of participants (22%, 102/459) were classified as robust (Fried score=0) at baseline, with the majority (62%, 285/459) classified as pre-frail (Fried score=1 or 2)”
Point 5 Line 279 you state: “We observed a significant association between co-morbidity and frailty, similar to that observed in previous work.” Please explain the co-morbidities being referred to and how these are related to the frailty assessment. Please also state what the previous work was, by whom and when.
Point 5a: You have not explained what the co-morbidity's are as requested in point 5
Response 5: Thank you for highlighting the needs for clarity. We have clarified the association between the number of co-morbidities and frailty, and added relevant reference: Mendonça N, Kingston A, Granic A, Jagger C. Protein intake and transitions between frailty states and to death in very old adults: the Newcastle 85+ study. Age Ageing. 2019;49(1):32-8.
Response 5a: It is the number of co-morbidities that was associated with frailty and we have amended the sentence.
We have also listed the conditions in the Methods section 2.3 Line 133 - 137.
Point 6 Based on the limitations in this study there is very little confidence in the outcomes as you have made substantial assumptions. Can you please explain these further if possible.
Point 6a With respect to point 6 the confidence levels to accept the results submitted have not been ratified or explained.
Response 6: Some of the key challenges with longitudinal studies is attrition and participants’ burden, particularly in older adults. We have applied a conservative assumption in managing the missing frailty items. Dietary intake is largely determined by our eating behavior which is quite stable throughout the life-course. Probable factors impacting eating habits include, but not limited to, changes in living situation, health status, and medication used which this age group likely to have adapted to (Bukman, A.J., Ronteltap, A. & Lebrun, M. Interpersonal determinants of eating behaviours in Dutch older adults living independently: a qualitative study. BMC Nutr 6, 55 (2020)). All statistical models have assumptions and multi-state models do a good job at modelling transitions between frailty states and death. However, we defined the transition probability as 0 for some transitions within non-adjacent states (e.g., robust to frail) because of non-convergence issues with the models.
Response 6a: We acknowledged the wide confidence intervals as a study limitation attributed to the low number of transitions/small sample size (Line 381-382 and 434-436)
